# Current Views on the Roles of *O*-Glycosylation in Controlling Notch-Ligand Interactions

**DOI:** 10.3390/biom11020309

**Published:** 2021-02-18

**Authors:** Wataru Saiki, Chenyu Ma, Tetsuya Okajima, Hideyuki Takeuchi

**Affiliations:** 1Department of Molecular Biochemistry, Nagoya University Graduate School of Medicine, Nagoya, Aichi 466-8550, Japan; saiki.wataru@e.mbox.nagoya-u.ac.jp (W.S.); machenyuyaoxue@yahoo.co.jp (C.M.); tokajima@med.nagoya-u.ac.jp (T.O.); 2Institute for Glyco-core Research (iGCORE), Nagoya University, Nagoya, Aichi 464-8601, Japan

**Keywords:** *O*-Glycosylation, Notch signaling, Notch receptors, Notch ligands, delta, serrate, EGF repeats, glycosyltransferase, mass spectrometry

## Abstract

The 100th anniversary of Notch discovery in *Drosophila* has recently passed. The Notch is evolutionarily conserved from *Drosophila* to humans. The discovery of human-specific Notch genes has led to a better understanding of Notch signaling in development and diseases and will continue to stimulate further research in the future. Notch receptors are responsible for cell-to-cell signaling. They are activated by cell-surface ligands located on adjacent cells. Notch activation plays an important role in determining the fate of cells, and dysregulation of Notch signaling results in numerous human diseases. Notch receptors are primarily activated by ligand binding. Many studies in various fields including genetics, developmental biology, biochemistry, and structural biology conducted over the past two decades have revealed that the activation of the Notch receptor is regulated by unique glycan modifications. Such modifications include *O*-fucose, *O*-glucose, and *O*-*N*-acetylglucosamine (GlcNAc) on epidermal growth factor-like (EGF) repeats located consecutively in the extracellular domain of Notch receptors. Being fine-tuned by glycans is an important property of Notch receptors. In this review article, we summarize the latest findings on the regulation of Notch activation by glycosylation and discuss future challenges.

## 1. Introduction

Notch signaling is involved in the notched wing phenotype in *Drosophila*, originally observed by John S. Dexter [1]. Thomas Hunt Morgan identified the mutant alleles in 1917 [2]. With the advent of molecular biology, the cloning and sequencing of the gene encoding *Drosophila* Notch receptor was undertaken by Spyros Artavanis-Tsakonas and Michael W. Young independently in the 1980s [3,4], which drove Notch research into the molecular era. The main components and steps of the Notch signaling pathway have been elucidated. The Notch signaling pathway is a pivotal regulator of cellular fate and plays a significant role in the development and tissue renewal in metazoans. In humans, dysregulation of Notch signaling leads to numerous diseases ranging from developmental syndromes to adult-onset disorders [5]. Mutations of Notch pathway components cause monogenic diseases such as Alagille syndrome and spondylocostal dysostosis. Tumorigenesis is also related to dysregulation of Notch signaling in many cellular contexts, in which the Notch signaling pathway act oncogenic function such as in T-ALL as well as the tumor suppressor function such as in the small cell lung cancer [6].

Notch signaling is highly conserved in metazoans, regulating multiple processes involved in the development of multicellular organisms, tissue homeostasis, and stem cell maintenance [7]. The canonical Notch pathway is activated via *trans*-interaction involving the interaction between the receptors and ligands expressed on juxtaposed cells, leading to signal transduction (Figure 1A). Mammals have four Notch receptors (NOTCH1-4), two types of canonical DSL (Delta/Serrate/LAG-2 family) ligands: three Delta-like ligands (DLL1, DLL3, and DLL4), and two Jagged ligands (JAG1 and JAG2). Both receptors and ligands are type-I transmembrane proteins. The extracellular domain (ECD) of DSL ligands contain a conserved *N*-terminal DSL domain and several epidermal growth factor-like (EGF) repeats, whereas the ECD of Notch receptors (NECD) consists of a series of EGF repeats and a negative regulatory region (NRR) (Figure 1B). As Notch receptors are translated, their ECDs undergo protein folding and glycosylation in the endoplasmic reticulum (ER). The cleavage of the NECD by furin-like convertase (S1 cleavage) occurs in the Golgi apparatus, and the receptor finally forms a non-covalently associated heterodimer. The EGF repeats, which are defined by the presence of six conserved cysteine residues forming three disulfide bonds modified with *O*-glycans at distinct sites (Figure 1C), participate in receptor-ligand interactions [8]. Endocytosis of the ligand on the signal-sending cell generates a pulling force that unfolds the NRR [9], exposing the ADAM10/17 metalloprotease cleavage site, which allows ADAM-mediated processing (S2 cleavage) (Figure 1A) [10]. Following S2 cleavage, the γ-secretase complex mediates the intra-transmembrane cleavage (S3 cleavage), releasing the intracellular domain of Notch receptors (NICD), which is subsequently translocated into the cell nucleus. The NICD interacts with the DNA-binding protein for J kappa recombinant signal binding protein (RBPjk) and co-activator mastermind-like (MAML) to form a transcriptional complex that activates the expression of downstream target genes [11,12].

A principal understanding of the interaction between Notch and its ligands was obtained through *Drosophila* studies. Early studies showed a genetic interaction between *Notch* and *Delta*, and subsequent molecular interactions between them were first demonstrated by cell-based aggregation assays using *Drosophila* S2 cells [13]. The same study suggested that the interaction between NOTCH and DELTA (DL) is calcium-dependent and mediated by the ECDs of both NOTCH and DL.

Fringe (FNG), a secreted protein, was initially identified as a critical regulator in *Drosophila* wing development. FNG is expressed in the dorsal, but not the ventral, part of wing imaginal discs and inhibits the responsiveness of NOTCH to SERRATE (SER) expressed in dorsal cells, but potentiates the NOTCH response to DL expressed in ventral cells [14]. This mechanism allows for the formation of the dorsal-ventral wing margin [15]. Initial observations proposed that secreted FNG inhibited SER by interacting with NOTCH [16]. However, in 2000, two independent studies revealed that FNG is a glycosyltransferase that catalyzes the addition of *N*-acetylglucosamine (GlcNAc) to *O*-fucose [17,18]. The modulation of Notch-ligand interactions is dependent on the glycan structure and not on the FNG protein itself [17,18].

The discovery of the glycosyltransferase activity of Fringe in 2000 demonstrated an association of the Notch signaling pathway with glycobiology [19]. For nearly two decades, many studies have revealed intriguing aspects of *O*-linked glycans on these EGF repeats in the regulation of Notch-ligand interactions [18,20,21,22,23,24,25]. In particular, *O*-fucose glycans directly participate in the binding of NOTCH1-DLL4 and NOTCH1-JAG1 and demonstrate critical functions [26,27,28]. Here, we summarize the structures of *O*-glycans attached to Notch receptors and discuss the latest findings on Notch-ligand interactions while focusing on *O*-linked glycans located on the EGF repeats.

## 2. Glycosylation on Notch/Recognition of EGF Repeats by Glycosyltransferases

### 2.1. Structural Classification of O-Glycans Is Critical to Elucidate Notch Receptor Functions

In 1985, the Artavanis-Tsakonas group reported a major embryonic Notch transcript that contains 36 tandemly arranged EGF repeats [30]. However, the function of these EGF repeats remained a mystery for a long time. In 1991, the Rebay group observed that EGF11 and 12 are both necessary for interaction with DL and constitute a SER binding domain [31]. In 2000, two unusual forms of *O*-linked glycans, namely, *O*-fucose and *O*-glucose, were initially described on the EGF repeats of mouse NOTCH1 [19]. Since then, studies have shown that NECDs are covered with multiple types of glycans, which can be classified as *N*-glycans attached to an asparagine residue, and *O*-glycans that are attached to serine or threonine residues. These *O*-glycans on EGF repeats of NECD can be subdivided into *O*-glucose, *O*-fucose, *O*-GlcNAc, and possibly *O*-xylose [32]. Each *O*-glycosylation reaction occurs at specific positions in properly folded EGF repeats on Notch receptors or DSL ligands via specific glycosyltransferases. These enzymes are localized in the ER [22,25,33,34,35,36,37]. A diagram of an EGF repeat modified with glycans at specific consensus sequences along with the responsible enzymes is shown in Figure 1C.

A summary of *O*-glycosylation observed on mouse NOTCH1 is shown in Figure 2A. Several semiquantitative analyses using mass spectrometry revealed variable stoichiometries at the *O*-glycosylation sites (Figure 2A) [25,28,38,39]. In each study, the soluble form of C-terminal Myc- and His_6_- tagged mouse NOTCH1 EGF repeats was transiently overexpressed in HEK293T cells and collected from the culture media (Figure 2B). These semiquantitative studies have helped us understand the substantial effect of *O*-glycosylation and its elongation at each predicted site. *O*-Glycans play critical and multiple roles in the activation of Notch receptors, including protein folding/stability, trafficking, and ligand binding. Not surprisingly, several human diseases are related to mutations in genes encoding Notch-modifying glycosyltransferases. For instance, nine missense mutations and one nonsense recessive variant in *POGLUT1* have been described in Limb-girdle muscular dystrophy (LGMD) R21 patients belonging to unrelated families from different countries [40,41]. Further details can be found in recent review articles reported by us and other groups [42,43,44,45].

### 2.2. Advances in the Understanding of the Recognition and Modification of Notch EGF Repeats by Enzymes

Currently, more than 110 families of glycosyltransferases have been reported in the carbohydrate-active enzyme database (CAZy) [46]. These glycosyltransferases catalyze the transfer of a monosaccharide from a donor substrate (commonly in the form of nucleoside diphosphate sugars, e.g., UDP-glucose and GDP-fucose) to specific acceptor substrates (e.g., EGF repeats), forming glycosidic bonds. Since the first X-ray structure for bacteriophage T4 glucosyltransferase was reported in 1994 [47], numerous crystal structures of glycosyltransferases alone or with substrates have been resolved [48]. Most of them adopt two predominant structure folds, namely, the GT-A fold, which consists of two closely abutting β/α/β Rossmann-like folds, or GT-B fold, which consists of two flexibly linked β/α/β Rossmann-like domains that face each other [49].

In the last few years, significant advances have been made in elucidating the structures of enzymes responsible for *O*-glycosylation of Notch receptors. X-ray crystal structures of the glycosyltransferases *Drosophila* Rumi and human POGLUT1 (GT-B fold, CAZy glycosyltransferases family 90), mouse XXYLT1 (GT-A fold, CAZy glycosyltransferases family 8), and human POFUT1 (GT-B fold, CAZy glycosyltransferase family 65) in complex with donor or acceptor substrates have been reported [50,51,52,53]. In general, for GT-B fold glycosyltransferases, the geometry of the cleft between donor binding subsites and acceptor binding subsites vary greatly among enzymes to accommodate different acceptors such as glycan and protein domains [48]. The structural data provide detailed mechanisms for the recognition of properly folded EGF repeats using these enzymes [51,52,53,54]. The large cleft of POFUT1 and POGLUT1 show high complementarity with the folded EGF acceptor substrates. For example, POGLUT1 recognizes common 3D features of the properly folded EGF repeats that have the characteristic kinked loop of an *O*-glucosylation consensus sequence as well as the conserved hydrophobic residue apart from the modification site in a primary sequence.

Typically, glycosyltransferases can be classified as retaining enzymes or inverting enzymes based on the anomeric linkage of a given sugar moiety in a donor nucleotide sugar and a product. A retaining enzyme catalyzes the transfer of a glycosyl group to an acceptor with retention of an anomeric configuration, while an inverting enzyme catalyzes the reaction with inversion of the anomeric configuration. The reaction mechanism of most inverting glycosyltransferases is generally considered as a single displacement S_N_2 mechanism; however, the retaining mechanism of glycosyltransferases is currently being debated [48,49]. Two mechanisms have been proposed. The double displacement mechanism was first suggested; however, since it lacked experimental evidence, in recent years some researchers proposed the S_N_i-like (I for internal return) front-face mechanism, which retains configuration by letting the acceptor nucleophile attack the donor anomeric carbon from the same side as the leaving group. In the analysis of wild-type XXYLT1, a transient Michaelis complex of three components was observed, allowing determination of the S_N_i-like retaining mechanism by XXYLT1 [50]. Furthermore, the fact that the EGF repeat undergoes a significant conformational change after the formation of the Xyl-Glc-EGF/XXYLT1 binary complex is achieved, which is presumably caused by the recognition and binding between both the disaccharide and EGF repeats with XXYLT1, suggests that EGF repeats have significant structural flexibility [50].

## 3. Notch-Ligand Interactions

### 3.1. Identification of Ligand-Binding Domain (LBD) in the Notch ECD

Early biochemical studies in *Drosophila* have enabled the molecular characterization of Notch. The molecular interaction of proteins encoded by *Notch* and *Delta* was observed by calcium-dependent aggregation of NOTCH-expressing cells and DL-expressing cells [13]. This interaction requires the ECD of NOTCH [13]. Later, a mutagenesis study demonstrated that the 11th and 12th EGF repeats (EGF11 and EGF12) from NOTCH are pivotal and sufficient for interaction with DL [31]. As a result, EGF11 and 12 are believed to be responsible for direct ligand binding (Figure 3). However, recent studies have provided evidence that confirmed a more extended LBD: EGF8-12 [27,28,55,56,57]. In 2012, Yamamoto et al. reported a unique *Drosophila* mutant called jigsaw, which was only activated by *Delta* but not by *Serrate* [55]. The *jigsaw* mutant carries one missense mutation on the evolutionarily conserved valine residue on EGF8 of NOTCH. This Val361 to Met mutation decreases the ability of NOTCH to bind to SER. Similar to *Drosophila* NOTCH, the single amino acid substitution of the conserved Val to Met on NOTCH2, also inhibits JAG1-induced NOTCH2 activation, while DLL1-induced NOTCH2 activation is retained. These findings provided new insights into Notch-ligand interactions and raised the possibility that not only EGF11 and EGF12 but also EGF8 participate in direct interaction with ligands, proposing an expansion of LBD. Later, Luca et al. resolved the X-ray structures of NOTCH1-DLL4 and NOTCH1-JAG1 complexes [26,27] and visualized the involvement of EGF8 in JAG1 binding. Consistently, mice lacking NOTCH1 EGF8-12 showed severe developmental defects that were indistinguishable from *Notch1* null mice [58]. Since NOTCH1 lacking the LBD was expressed on the cell surface [58], this study further supports that EGF8-12 is important for the binding of NOTCH1 to the ligands.

### 3.2. Amino Acids and Sugars Directly Involved in Notch-Ligand Binding

Some studies have employed mutagenesis of residues in the Notch LBD to determine which residues are involved in ligand binding. The key residues discussed in this section are shown in Figure 3. The Handford group provided evidence showing that the binding of calcium to NOTCH1 EGF12 is needed for the binding of DLL1. The Asp469Gly-mutated NOTCH1 EGF11-13 fragments cannot bind to DLL1-expressing cells [59]. They also mutated in the conserved residues of EGF12 and other residues surrounding them and found critical residues for JAG1 binding such as Leu468, Glu473, Gln475, Ile477, and Met479 [60]. Ala substitution at corresponding residues on *Drosophila* NOTCH also confirmed the involvement of these residues in Notch-ligand interactions [61].

In the co-crystal structure of NOTCH1 EGF11-13 and DLL4 SLP N-EGF1 (SLP is an affinity maturated mutant), two primary sites for the interaction were observed [26]. The critical amino acids are shown in Figure 3. Site 1 is located between EGF12 of NOTCH1 and the MNNL domain of DLL4. At this site, Leu468, Asp469, and Ile477 directly interact with the residues from DLL4. As mentioned above, mutations in these residues lead to a loss of DLL1- or JAG1-binding [59,60]. The *O*-linked fucose at Thr466 also significantly contributes to binding at site 1 by interacting with His64, Tyr65, and Thr114 from DLL4. Site 2 is located between EGF11 of NOTCH1 and the DSL domain of DLL4. Residues such as Glu415, Pro422, Phe436, Glu450, and Asp452 in EGF11 (Figure 3) form a binding interface for DSL. Arg448 is deeply buried in the groove of the DSL. The *O*-linked glucose at Ser435 also contributes to binding at site 2 by interacting with Asp218 and Gln219 from DLL4.

The structure of the NOTCH1 EGF8-12 and JAG1 N-EGF3 complex revealed additional interactions of EGF10, 9, and 8 observed in NOTCH1 with EGF1, 2, and 3 observed in JAG1 [27]. Previously, the interface between NOTCH1 EGF9 and 10 was shown to possess flexible mobility [62]. In the NOTCH1-JAG1 complex structure, bending between EGF9 and 10 on the other side of the JAG1 binding surface, was observed [27]. JAG1 was also bent to fill the gap created by NOTCH1 EGF9 and 10. Interestingly, residues from JAG1, such as Tyr255, which are located at this interface are not conserved in DLL1 and DLL4, suggesting that the binding architecture of JAG1 to NOTCH1 EGF9 and 10 is distinct and not the same as that of DLL1 and DLL4 [27]. The addition of EGF10 to NOTCH1 EGF11-13 fragments decreased the binding affinity toward DLL1, which was restored via mutagenesis of the calcium-binding motif on EGF11 [59] such that EGF10 may interfere with binding with Delta ligands but not with JAG1. The *O*-fucose on Thr311 of NOTCH1 EGF8 and Val324 directly contact JAG1 at the interface between NOTCH1 EGF8 and JAG1 EGF3. The *O*-fucose residue at Thr311 of NOTCH1 interacts with Asn298 from JAG1. Val324 is a conserved residue whose substitution with Met in *Drosophila* Notch and mammalian NOTCH2 results in loss of JAG1/SER-induced activation; however, DLL1/DL-induced activation remains unchanged [55].

### 3.3. Roles of O-Glycans Located on LBD in Ligand Interactions in Drosophila Notch and Mammalian NOTCH1

The direct involvement of *O*-fucose located on NOTCH1 EGF12 in DLL4 interaction explains the results of many studies that have shown a significant role for *O*-fucose located on EGF12 under in vitro and in vivo conditions in various organisms such as *Drosophila* and mammals.

#### 3.3.1. *O*-Fucose Glycans on *Drosophila* Notch LBD

In *Drosophila* NOTCH, *O*-fucosylation occurs within the consensus sequence of EGF repeats with high stoichiometry [63]. Genetic deletion or knockdown of *Ofut1* results in the loss of Notch signaling, which strongly suggests that *O*-fucosylation plays a central role in Notch signaling [64,65]. In addition, global elimination of *O*-fucose on NOTCH reduces its binding to both Delta and Serrate ligands [20] and overexpression of *Ofut1* for the production of soluble Notch fragments significantly enhances the binding of NOTCH to SER [20]. This suggests that *O*-fucose monosaccharides play a distinct role in Notch and ligand interactions. Recently, the Haltiwanger and Jafar-Nejad groups demonstrated the critical role of *O*-fucose monosaccharides located on EGF8 and 9 in interactions with the Serrate ligand [57]. To date, no reports of co-crystal structures of the *Drosophila* Notch receptor and ligand have been published. However, based on the structure of NOTCH1 and Serrate homolog JAG1 [27], it is plausible that *O*-fucose monosaccharide located on EGF8 is important. *O*-Fucose on EGF9 is not captured in the same complex, so additional visualization is required. Interestingly, mutation of the *O*-fucose site on EGF12 enhances NOTCH-SER binding [57,66]. The molecular basis for the inhibition of SER binding by *O*-fucose on EGF12 should be addressed in future studies. In addition, Pandey et al. demonstrated that *O*-fucose monosaccharides on EGF12 and EGF9, but not EGF8, are important for DL-induced Notch signaling during *Drosophila* embryonic neurogenesis. In particular, the sole mutation of the *O*-fucose site on EGF12 results in a neurogenic phenotype when expressed at near-endogenous levels, suggesting that *O*-fucose on EGF12 is necessary for DL-mediated Notch activation in vivo [57]. This study underscores the importance of setting the appropriate Notch expression level comparable to that in vivo to determine the effects modulated by *O*-linked glycans precisely. Overall, *O*-fucose monosaccharides on Notch LBD are critical, but their contributions differ with each ligand interaction.

*O*-Fucose monosaccharides can be elongated with GlcNAc residues using FNG. This elongation further modulates the activity of NOTCH. In *Drosophila*, additional elongation to GlcNAcylated *O*-fucose on NOTCH has not been reported [63]. FNG modification of NOTCH blocks binding to SER while enhancing binding to DL. In the search for the responsible site(s), Lei et al. showed that *O*-fucose on EGF12 is one of the multiple sites that are pivotal for ligand discrimination [66]. However, although regions containing EGF6-9 are predicted to be important [67], the contribution of other sites remains unclear. A more recent study showed that mutation of the *O*-fucose site on EGF8 diminishes the inhibitory effect of Fringe on SER binding [57]. GlcNAc-extended *O*-fucose glycans on both EGF8 and 12 are important for interaction with DL [57].

#### 3.3.2. Critical Roles of *O*-Fucosylation in the LBD of Mammalian Notch

*O*-Fucosylation by POFUT1 is also essential for Notch activation in mammals [68]. In mouse NOTCH1, site-directed mutagenesis of the *O*-fucose site on EGF8 or 12 decreases NOTCH1/DLL1 binding and activation in the absence of all three Fringes [28]. However, mutations on EGF8 and 9, but not 12, decrease NOTCH1/JAG1 binding and activation [27,28]. The combination of mutations on both EGF8 and 12 remarkably decreases the binding to DLL1 and JAG1 and their activation, suggesting that both sites are redundant to a certain degree. Taken together, similar to flies, *O*-fucose monosaccharides on the LBD are essential but contribute differentially toward ligand binding. Using mice model under in vivo conditions, elimination of *O*-fucose on EGF12 in the 129S2/SvPasCrl background retains viability and fertility and affects T cell development in the thymus [69]. However, in the C57BL/6J background, the same mutation results in embryonic lethality, suggesting the presence of a genetic modifier [70].

*O*-Fucose located on mouse NOTCH1 is also extended with GlcNAc residues via Lunatic Fringe (LFNG), Manic Fringe (MFNG), or Radical Fringe (RFNG); however, modification by MFNG is poor [28]. Unlike the *Drosophila* SER ligand, elongation of *O*-fucose enhances NOTCH1-JAG1 binding while decreasing NOTCH1 activation induced by JAG1 (Table 1) [28]. Using the specificity of each Fringe and introducing the mutations on *O*-fucosylation sites, GlcNAc-elongation on EGF12 enhanced both NOTCH1-DLL1 and NOTCH1-JAG1 binding [28,71], while elongation on EGF8 only enhanced NOTCH1-DLL1 binding [28]. Interestingly, elongation of *O*-fucose on EGF12 does not enhance NOTCH1-DLL4 binding to the same extent as NOTCH1-DLL1 [71]. The distinct properties of DLL1 and DLL4 described elsewhere [56,72,73] may be attributed to the effect of sugar-modification patterns. In the crystal structures of NOTCH1, the GlcNAc residue of *O*-fucose on EGF12 protrudes from the backbone and provides an expanded ligand-binding surface [26,71]. The GlcNAc residue modeled onto the co-crystallized structures of the NOTCH1-DLL4 complex appears to interact with Met479 from NOTCH1 and with ligands directly [26]. GlcNAcylated *O*-fucose glycans can be further elongated by galactose and sialic acids [19,28,74]. Elongation by galactose and sialic acids on EGF12 does not enhance NOTCH1-DLL1 or NOTCH1-JAG1 binding [71]. In addition, in the presence of only RFNG, *O*-fucose on EGF12 remains a disaccharide; however, enhanced binding is still observed [28]. Therefore, consistent with the results of the study conducted in *Drosophila* [21], *O*-fucose disaccharide on EGF12 sufficiently enhances the binding affinity with ligands. In contrast, the Fringe effect for inhibiting JAG1-induced NOTCH1 activation requires the addition of galactose to GlcNAc-extended *O*-fucose [29,75]. In addition, the effect of LFNG, but not MFNG, on the enhancement of NOTCH1-DLL1 binding was diminished when galactosyl extension was eliminated [75]. The molecular basis for the effects of the elongation of *O*-fucose glycans by galactose and sialic acids remains unclear; however, they are presumably attributed to EGF repeats outside the LBD.

Recently, Schneider et al. reported that fucose analogs (6-alkynyl and 6-alkenyl fucoses) are transferred to NOTCH EGF repeats and specifically inhibit binding to and activation by DLL1 or DLL4, but not JAG1, to NOTCH1 and NOTCH2 [76]. Interestingly, mutation of *O*-fucose sites on EGF8, but not EGF12, diminishes the inhibitory effect of the fucose analogs, indicating that *O*-fucose on EGF8 plays an important role in the direct interaction with mammalian Delta ligands. Structural models of the interaction of NOTCH1 and JAG1, in which the fucose analogs are incorporated, further support the lack of steric clash of the fucose analog on EGF8 or 12 with JAG1. Co-crystallization analysis of NOTCH1, including EGF8 and EGF12 of NOTCH1 and extended domains of DLL1 or DLL4, needs to be performed to determine whether there is a clash with Delta ligands.

#### 3.3.3. *O*-Glucose Glycans in the LBD of Notch

In contrast to *O*-fucose, classical *O*-glucose catalyzed by POGLUT1 on NOTCH1 EGF12 is located on the opposite surface of the EGF repeat from the *O*-fucose and is not directly involved in binding [26,27,77] (Figure 3). In *Drosophila*, unlike *O*-fucose sites, Ser to Ala substitution at the *O*-glucosylation site on EGF12 does not show a *Notch* phenotype [78]. In addition, elongation of *O*-glucose with xylose residues on EGF12 and 13 from NOTCH1 does not alter the binding affinity of human NOTCH1EGF11-13 to DLL1, DLL4, or JAG1-expressing cells [71]. Although the knockdown of *Poglut1* in C2C12 cells decreases *O*-glucosylation at specific sites and NOTCH1 activation, DLL1- and JAG1-binding are not altered [79]. Taken together, POGLUT1-synthesized *O*-glucose glycans do not directly participate in ligand interactions.

The recently identified *O*-glucose on EGF11 added by POGLUT2 or 3 is present on the surface that binds with DLL4 and interacts with Asp218 and Gln219 of DLL4 [26]. However, elimination of this *O*-glucose on EGF11 does not affect NOTCH1-DLL1 binding or DLL1-induced NOTCH1 activation [25]. This is consistent with the relatively weaker interaction of *O*-glucose at Ser435 and DLL4 observed in the NOTCH1-DLL4 complex [26]. Intriguingly, in combination with an *O*-fucose site mutation on EGF8, elimination of the *O*-glucose on EGF11 significantly decreased NOTCH1 cell surface expression compared to the EGF8 mutation alone. The combination of the EGF11 mutant with *O*-fucose site mutation on EGF12 also decreases DLL1-induced activation while maintaining DLL1-binding (Table 1) [25]. These results suggest that the interaction between *O*-glucose at Ser435 and DLL1 is important to a certain degree.

#### 3.3.4. *O*-GlcNAc Glycans in the LBD of Notch

EOGT-mediated *O*-GlcNAcylation promotes NOTCH1 and DLL1/4 binding with no effect on JAG binding [24,80]. Consensus sequences for *O*-GlcNAcylation were observed on EGF8, 9, 10, and 11 within the LBD of mouse NOTCH1. To date, mass spectrometric analysis has revealed a low level of *O*-GlcNAcylation on EGF10 and 11, whereas *O*-GlcNAc glycoforms on EGF8 and 9 have not yet been determined [39]. In the NOTCH1 and JAG1 complex structures, *O*-GlcNAc is depicted on EGF11 but not on the JAG1 binding surface [27]. An NMR study demonstrated a hydrophobic interaction of *O*-GlcNAc at Ser445 of EGF11 with Ile451. *O*-GlcNAc at Ser405 of EGF10 interacts with Gln411 [81] (Figure 3). Both Ile451 and Gln411 are located at the hinge region of two sequential EGF repeats. Hence, the authors proposed that *O*-GlcNAc as well as calcium ions regulate the flexibility and rigidity of the connection of NOTCH1 EGF repeats [81]. Very recently, we demonstrated that similar to *O*-fucose and *O*-glucose [82], *O*-GlcNAc enhances the stability of a single EGF repeat, *Drosophila* NOTCH EGF20 [83]. Further insights are required into how *O*-GlcNAc alters the rigidity of sequential EGF repeats, whether *O*-GlcNAcylation and calcium ions affect each other, and if either of these affects the activation of Notch.

### 3.4. Calcium-Binding of EGF Repeats

Notch binding to ligands is dependent on calcium [59,75,84]. EGF repeats are known to interact with calcium ions. The consensus sequence for calcium binding is predicted to be D/N-x-D/N-E/Q-Xm-D/N*-Xn-Y/F (* indicates the site for β-hydroxylation) [62,85,86,87]. In mouse NOTCH1, 20 out of 36 EGF repeats possess a calcium-binding motif (Figure 2). The *N^M1^* mutation in *Drosophila* demonstrated a Glu to Val substitution located in NOTCH EGF12 [88]. *N^M1^* mutants behave as both dominant-negative and loss-of-function mutations. This Glu residue is located within the consensus sequence and participates in calcium-binding [89], suggesting a critical role of calcium-binding in the Notch LBD for its activity.

An Asp469Gly mutation disrupting the calcium-binding consensus sequence of EGF12 of *NOTCH1* was identified in a cutaneous squamous cell carcinoma cell lines [90]. A tumor suppressor role of *NOTCH1* has been proposed. NOTCH1 with Asp469Gly mutation was not activated by JAG2 [90]. The Asp469Gly mutation depletes calcium-binding to NOTCH1 EGF12. Flow cytometric assays conducted using NOTCH1 fragments and DLL1-expressing cells showed that Asp469Gly does not bind to DLL1 [59]. The same study also demonstrated that calcium-binding to EGF11 and EGF13, but not EGF12, is dispensable for DLL1 binding. Recent structural studies have shown that Asp469 is also directly involved in the interaction with DLL4 and JAG1 [26,27]. Hence, both calcium ions and an Asp residue at 469 are important for ligand binding. In addition, Asp469 is predicted to be β-hydroxylated; however, the role of β-hydroxylation in the function of Notch remains unclear. Mutations disrupting the calcium-binding consensus sequence increase the proteolytic susceptibility of NOTCH1 fragments [59] and removal of calcium ions reduces the stability of human coagulation factor IX EGF1 in the presence of a reducing agent, dithiothreitol [82]. Therefore, calcium ions may be necessary for the formation of rigidity, which could be important for the generation of an efficient pulling force to initiate Notch proteolytic activation. In addition, calcium-binding affinity is different at each site within NOTCH1 [62], and it would be interesting to examine whether glycosylation, as well as an amino acid sequence surrounding the calcium-binding sites, modulate the affinity for calcium.

### 3.5. Structure and Function of O-Glycosylation Outside of the LBD

#### 3.5.1. Importance of EGF Repeats Outside of the LBD in Ligand Interaction

Other regions outside the LBD (EGF8-12) are also known to affect Notch ligand interactions. Andrawes et al. concluded that EGF6-15 of NOTCH1 is sufficient for ligand stimulation [56]. Their study demonstrated that the deletion of EGF1-5 or EGF16-36 does not alter either DLL1- or DLL4-induced activation, suggesting that EGF1-5 and 16–36 are not necessary. However, deletion of EGF6 and 7 decreased DLL1-induced activation, while DLL4-induced activation was retained. The X-ray structure of EGF4-7 revealed a bent structure in the hinge of EGF5 and 6 [62]. This may be relevant to the lack of requirement for EGF1-5. Prior data also showed that *Drosophila* NOTCH EGF1-5 is dispensable for ligand binding [67]. However, EGF24-26 in *Drosophila* NOTCH was shown to be required for interaction with DL and SER [91]. Truncation of EGF25-36 considerably impairs the interaction [67]. The addition of EGF14 to human NOTCH1 EGF11-13 constructs did not enhance binding to DLL1 [59]. Interestingly, a NOTCH1 decoy consisting of EGF1-24 has a stronger effect on the inhibition of JAG1-induced NOTCH1 activation than an EGF1-13 decoy, suggesting that EGF14-24 may increase the affinity for JAG1 [92].

Early studies on *Abruptex* mutants, which lead to hyperactivation of *Notch* in *Drosophila*, proposed a potential role for the *Abruptex* region (EGF24-29) in the dimerization of NOTCH [93]. *Drosophila* NOTCH appears to form a dimer via an intermolecular disulfide bond [94]. In vitro binding assays using shortened fragments from NOTCH EGF repeats demonstrated the presence of interaction between EGF11-14 and EGF22-27 and competition between EGF22-27 and Delta ligand for EGF11-14 [95]. Similarly, NOTCH1 can form dimers at the cell surface. However, this dimerization appears to be unnecessary for the activation [96]. A recent study showed that the clustering of Notch receptor-ligand impacts the distinct mode of DLL1-induced NOTCH1 activation in pulses [97]. Hence, although oligomerization is not fully required, it may provide a distinguishable feature of the activation patterns of each receptor-ligand pair.

#### 3.5.2. Roles of *O*-glycosylation Observed on EGF1-7

Elongation of *O*-fucose on NOTCH1 EGF6 was shown to inhibit JAG1-induced NOTCH1 activation [28]. It is unclear whether this glycan directly interacts with JAG1. Given that the elimination of this site by introducing Thr to Val mutation does not alter NOTCH1-JAG1 binding, it was proposed that Fringe-mediated elongation of *O*-fucose on EGF6 induces a conformational change, thereby inhibiting NOTCH1 activation.

Recently, human-specific genes belonging to the *NOTCH2NL* family have been identified, which play an important role in the human cortical expansion [98,99,100]. The proteins encoded by this gene family contain NOTCH2 EGF1-6. The EGF repeats encoded in *NOTCH2NLB* follow a signal peptide and are followed by a short *C*-terminal sequence. The function of NOTCH2NLB requires EGF repeats; however, the molecular mechanisms proposed by the two groups are different. Suzuki et al. reported that NOTCH2NLB inhibits DLL1 expression in the same cell and renders the cell more susceptible to ligand stimulation from other cells [98]. In contrast, Fiddes et al. demonstrated that NOTCH2NLB directly binds to NOTCH in the same cell and enhances its activation [99]. It is intriguing that in the original *NOTCH2*, the intermolecular interaction of NOTCH2 EGF1-6 with Notch ligands or intramolecular interaction NOTCH2 itself may take place. It is worth noting that among the four *NOTCH2NL* paralogs, the Thr residue for *O*-fucosylation on EGF5 is substituted with Ile in *NOTCH2NLA* and *NOTCH2NLB*. It is not known whether these proteins are glycosylated.

#### 3.5.3. Roles of *O*-glycosylation Observed on EGF13-36

The function of *O*-glycans in EGF13-36 also remains elusive. As summarized in Figure 4A, in mouse NOTCH1, the Thr to Ala mutation at an *O*-fucose site on EGF26 leads to the hyperactivation of NOTCH1 by JAG1 and DLL1 in the cell-based Notch reporter assay using COS-7 cells that express endogenous Fringe [101]. However, upon using NIH3T3 cells that do not express endogenous Fringe, both Thr to Ala and Thr to Val mutations of the *O*-fucose site on EGF26 do not show the same activating effect [28]. Furthermore, *O*-glucose on EGF28 appears to have a remarkable contribution to NOTCH1 activation. A Ser to Ala mutation at this site makes NOTCH1 unresponsive to DLL1 but does not affect the responsiveness to JAG1 [102]. In addition, as with *O*-fucose on EGF6, Fringe-mediated elongation of *O*-fucose on EGF36 does not alter the affinity of NOTCH1 and JAG1 but inhibits JAG1-induced NOTCH1 activation [28].

In *Drosophila*, a mutant called split that introduces an *O*-fucosylation site on EGF14 shows enhanced Notch activation in specific cells (Figure 4B) [103]. Likewise, xylosyl-extension of *O*-glucose glycans on EGF16-20 negatively regulates DL-induced Notch signaling in specific contexts (Figure 4B) [104,105,106].

### 3.6. Role of O-Glycans in Cis- and Trans-Interactions

Typically, four modes of ligand-induced Notch signal transduction are possible: adjacent-cell (*trans*)-activation, *trans*-inhibition, same-cell (*cis*)-inhibition, and *cis*-activation [107]. *Cis*-inhibition plays an important role in the development of *Drosophila* and other organisms [8,108]. Based on structural and mutagenesis experiments, *cis*-interaction is predicted to occur on the same surface of Notch as a *trans*-interaction [61]. Both LFNG and MFNG inhibited JAG1-induced *trans*-NOTCH1 activation, while RFNG enhanced it [28]. Fringe proteins appear to regulate *cis*-interactions in the same way as *trans*-interactions. LFNG and MFNG expression reduces the *cis*-interaction of NOTCH1 and JAG1. However, RFNG expression enhances their *cis*-interaction [23].

A recent single-cell imaging study reported the presence of *cis*-activation as the typical mode of Notch signaling between multiple Notch-ligand pairs in diverse cell types [107]. Interestingly, RFNG enhanced the DLL1-mediated *cis*-activation of NOTCH1, suggesting that *cis*-activation is also sensitive to the modification patterns of *O*-glycans. Co-expression of RFNG did not enhance DLL4-induced *cis*-activation of NOTCH1 [107]. Consistent with this result, recent studies have demonstrated that none of the mammalian Fringes can enhance DLL4-mediated NOTCH1 *trans*-activation in NIH3T3 cells [109]. In addition, NOTCH2 shows stronger *cis*-activation than NOTCH1 and seems to lack *cis*-inhibition [107]. Intriguingly, the different amplitudes of DLL1- or DLL4-induced NOTCH2 activation described elsewhere [72,109] were not observed in the *cis*-activation [107].

### 3.7. O-Glycosylation on Notch Ligands

The specific roles of *O*-glycans in Notch ligands remain unclear. *O*-Glycans on Notch ligands have been identified through structural studies and mass spectrometric analyses [26,27,110]. The *O*-fucose on JAG1 EGF3 directly interacts with His313 of NOTCH1 EGF8 as demonstrated by a co-crystal structure with NOTCH1. For the activity of mouse DLL1, *O*-fucosylation is not required [111]. In contrast, *O*-fucosylation is indispensable for the function of mouse DLL3 [112]. Regarding *O*-glucosylation, JAG1 has four POGLUT1-mediated *O*-glucosylation sites out of 16 EGF repeats in its ECD, and each site is efficiently *O*-glucosylated [113]. Reducing the gene dosage of *Poglut1* in mice increases the expression level of JAG1 and the soluble form of JAG1 and enhances the function of JAG1 [113]. It is not clear whether *O*-glucose glycans on JAG1 impact its binding affinity toward Notch receptors.

Curiously, wild-type NOTCH1 EGF1-14 fragments do not bind to DLL4 expressed in *S. cerevisiae* yeast, which does not express endogenous *O*-glycosyltransferases [26]. *O*-Glycans on Notch ligands may also contribute to the fine-tuning of Notch/ligand affinity. Additional experiments are required to confirm this.

## 4. Perspective

Here, we summarized the current understanding of how Notch and its ligand interactions are regulated by *O*-glycosylation. The function of the *O*-glycans is site-specific, in other words, which EGF repeats are attached to the glycans is also important for the function of the *O*-glycans. In particular, *O*-fucose glycans appear to function both directly and indirectly in the recognition of receptors by ligands. To clarify the molecular mechanism by which *O*-glycosylation of EGF repeats other than the ligand-binding region is involved in the interaction, it will be necessary to reveal the overall structure of the receptor including the *O*-glycans.

To date, many studies have demonstrated the specific roles of *O*-linked glycans on EGF repeats observed on mammalian NOTCH1 and *Drosophila* NOTCH; however, few have focused on NOTCH2-4. The distinct roles of each receptor-ligand pair have been described in several organisms [114]. *O*-Fucosylation by POFUT1 is also predicted to be critical for NOTCH2 [115]. However, unlike NOTCH1, one report showed that LFNG enhances the activation of both NOTCH2/JAG1 and NOTCH2/DLL1 signaling [116]. Another study reported that LFNG directly decreases NOTCH2/JAG1 binding, which is different from NOTCH1 [117]. Very recently, Kakuda et al. demonstrated that fringe effects on NOTCH2 are different from those on NOTCH1 [109]. However, many questions remain unanswered. *O*-Glucosylation catalyzed by POGLUT1 is not required for the function of NOTCH2 in the context of intrahepatic bile duct development in mice [113,118]. Given that *Poglut1*-knockdown in Neuro2A cells showed a clear NOTCH1-dependent axon phenotype [79,119], the requirements of *O*-glucosylation may be different in each cellular context and Notch paralog. Current studies have demonstrated that the distinct properties of NOTCH1 and NOTCH2 strongly depend on their ECDs [120,121]. In addition, in vitro analyses showed that the preference of NOTCH1 toward DLL4 and NOTCH2 toward DLL1 can be observed using small fragments containing EGF1-12 derived from mouse NOTCH1 and NOTCH2 [122], or EGF6-15 from human NOTCH1 and EGF1-15 from human NOTCH2 [72]. Similar to NOTCH2, the affinity of NOTCH3 toward DLL4 was lower than that of NOTCH1 [123]. Although their amino acid sequences and structures are quite similar [124], different patterns and elongation of *O*-glycosylation may make each paralog more distinct. For instance, two recently identified protein *O*-glucosyltransferases, namely, POGLUT2 and POGLUT3, do not modify NOTCH2 [25]. More detailed studies need to be performed to unveil how these unique *O*-linked glycans contribute to the modulation of the paralog-specific Notch activities and thereby regulate the development and homeostasis in vivo.

Finally, enhanced or reduced Notch signaling can result in a variety of diseases, including liver diseases and cancers [6,125]. In addition, inhibition or activation of specific Notch receptors and ligand axes have been proposed as therapeutics in other diseases such as nephrosis and graft-versus-host disease [126,127,128,129,130,131,132,133]. A recent study by Haltiwanger and Wu labs proposed an innovative way to specifically inhibit Delta ligand-induced Notch activities using fucose-analogs [76]. As it becomes more evident that *O*-glycans play a significant role in the regulation of the overall Notch signaling pathways, particularly Notch-ligand interactions, we need to take advantage of the modulation of Notch signaling by developing therapeutics that alter Notch glycosylation in specific contexts.

## Figures and Tables

**Figure 1 biomolecules-11-00309-f001:**
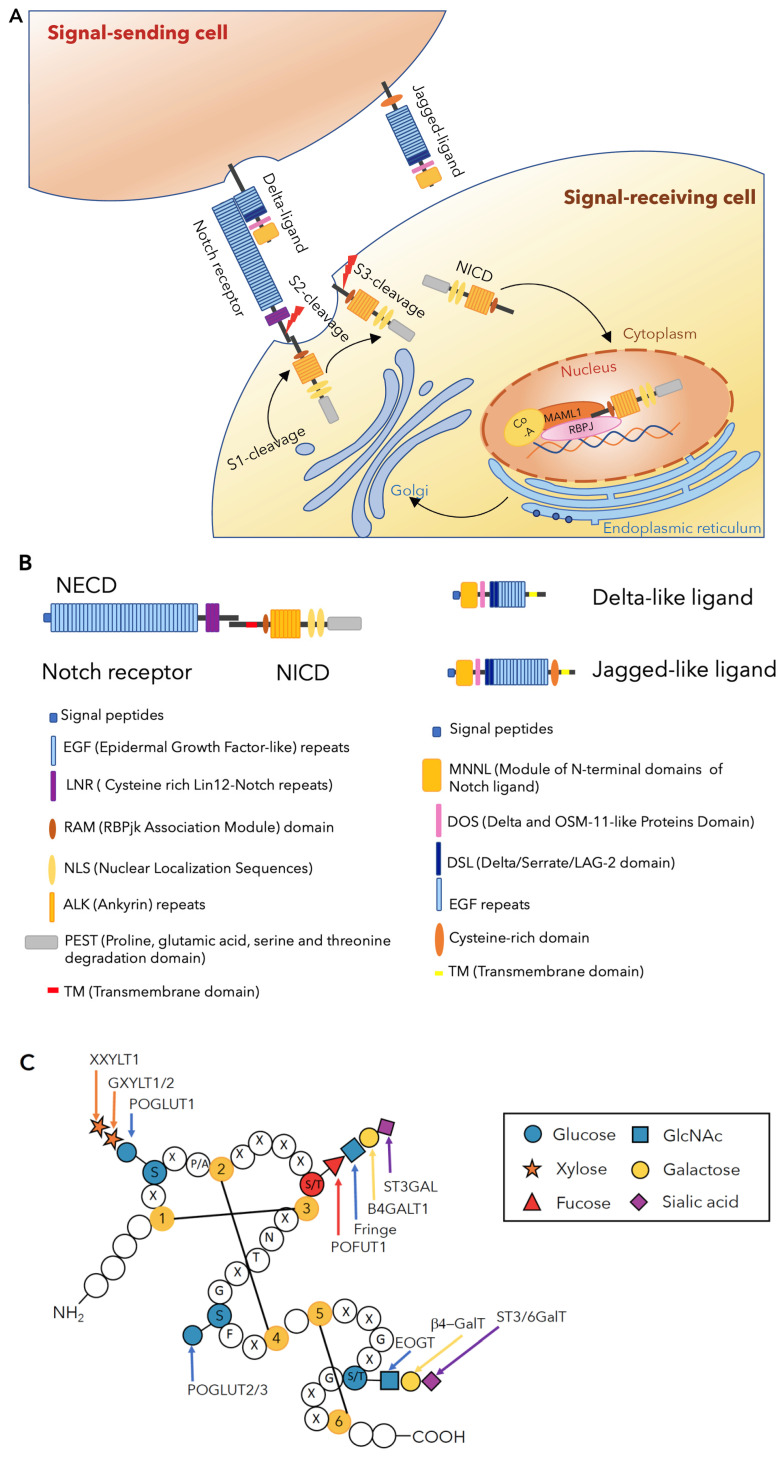
Molecular and structural basics of Notch receptors and activation. (**A**) The signal transduction process of Notch signaling. After the S1-cleavage during transport to the cell surface, the Notch receptor is expressed on the cell surface as a heterodimer. Upon Notch receptor-ligand binding, a pulling force is generated by endocytosis of the ligand, exposing the S2 cleavage site and followed rapidly by S3 cleavage. This releases the NICD that is translocated to the cell nucleus where it binds with several co-activators, thereby forming a transcriptional complex to activate the transcription of a downstream target. (**B**) The main components of the Notch receptors and ligands. In mammals, the extracellular domain of Notch receptors (NECD) consists of 29-36 EGF repeats, followed by three LNRs that are a part of the negative regulatory region (NRR). Mammalian Delta-like ligands and Jagged ligands belong to the DSL ligand family that is characterized by the presence of DSL (Delta/Serrate/LAG-2 domain). Among the three Delta-like ligands, DLL1 and DLL4 participate in Notch trans-activation. Multiple EGF repeats are also present in the DSL ligands. (**C**) Schematic diagram of a single EGF repeat, *O*-glycan structures, and glycosyltransferases that catalyze the modifications. An EGF repeat is defined by six conserved cysteine residues forming three disulfide bonds. The consensus sequence of the *O*-fucose modification is defined as C^2^X_4-5_(S/T)C^3^ (Amino acids that undergo glycosylation are underlined.). *O*-Fucose can be extended to GlcNAcβ1-3Fuc by Golgi-localized Fringe family glycosyltransferases. A sole Fringe is observed in *Drosophila*, contrary to the three Fringes observed in mammals, Lunatic Fringe (LFNG), Manic Fringe (MFNG), and Radical Fringe (RFNG). *O*-Fucose disaccharides can be elongated to a galactose-GlcNAc-Fuc trisaccharide by B4GALT1, and further to a tetrasaccharide by ST3GAL in mammals [17,29]. *O*-Glucose exists within the consensus sequence C^1^XSX(P/A)C^2^. Two homologs of POGLUT1, namely, POGLUT2 and POGLUT3, add *O*-glucose to the serine residue between the third and fourth cysteine on EGF11 of NOTCH1 and EGF10 of NOTCH3. *O*-GlcNAc occurs at the serine or threonine residue within the consensus sequence C^5^XXGX(S/T)GXXC^6^ in mammals, mediated by EOGT. The elongation of *O*-GlcNAc by galactose and sialic acid has been reported in mammalian cells.

**Figure 2 biomolecules-11-00309-f002:**
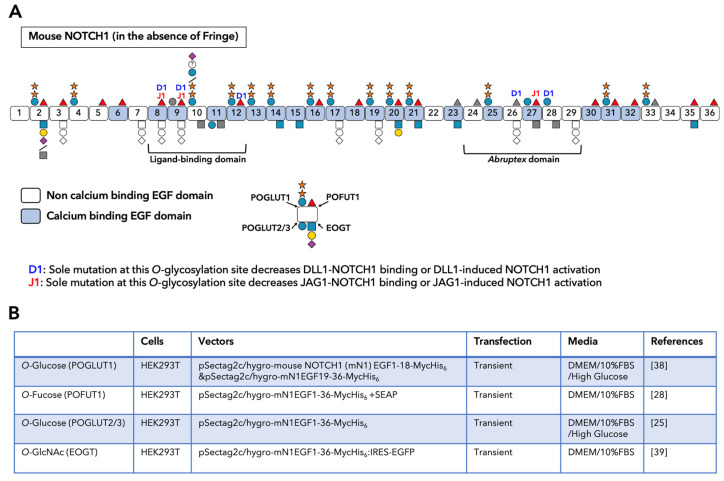
Schematic representation of mouse NOTCH1 and its *O*-glycan modifications. (**A**) The glycoforms of most of the predicted *O*-glycosylation sites in the absence of Fringes. The most abundant glycoforms are depicted for each site. Grey-colored sugar indicates that the predicted site appears to be unmodified, and white-colored sugar indicates that data for that site are not yet available. Sites where the elimination of *O*-glycosylation affects Notch-ligand binding or Notch activation are indicated: D1, affects DLL1/NOTCH1 interaction or DLL1-induced NOTCH1 activation; J1 affects JAG1/NOTCH1 interaction or JAG1-induced NOTCH1 activation. Sugar symbols are the same as in Figure 1C. (**B**) Studies that presented the mass spectrometric analysis of each type of *O*-linked glycans located on mouse NOTCH1 (mN1).

**Figure 3 biomolecules-11-00309-f003:**
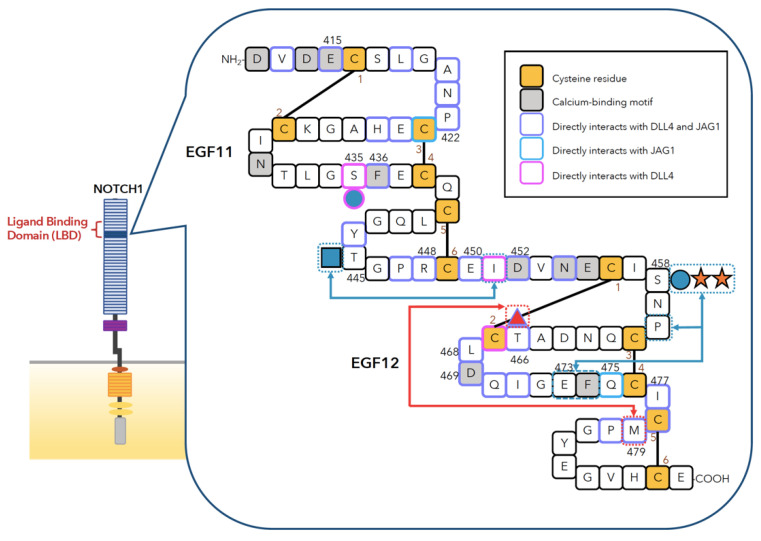
Schematic representation of residues in human NOTCH1 EGF11 and 12 interaction with ligands. Cysteine residues, residues within the calcium-binding motif, residues or *O*-glycans that directly interact with JAG1, DLL4, or both are filled with colors or indicated with colored enclosing lines, as shown in the upper right. Arrows show the intramolecular interactions between the *O*-glycans and amino acid residues. The sugar symbols are the same as those observed in Figure 1C.

**Figure 4 biomolecules-11-00309-f004:**
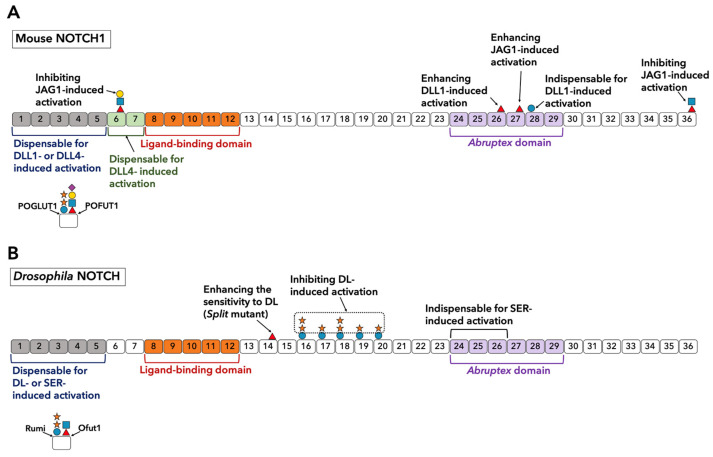
Some EGF repeats outside the LBD and *O*-glycans on them affect the ligand-induced Notch activation. (**A**) The role of EGF repeats outside LBD and *O*-glycosylation on them for the activation of mouse Notch. EGF1-5 are not required for the activation of NOTCH1 induced by DLL1 and DLL4. EGF6 and 7 are dispensable for the DLL4-induced activation; however, they are important for the DLL1-induced activation [56]. The elongated *O*-fucose glycans on EGF6 and 36 inhibits the JAG1-induced activation [28]. *O*-Fucose glycan on EGF26 is important for the DLL1-induced activation, while *O*-fucose on EGF27 is important for the JAG1-induced activation [28]. *O*-Glucose glycan on EGF28 is required for the DLL1-induced activation [102]. Only the key sugar residues described in the main text are depicted. Sugar symbols are the same as in Figure 1C. (**B**) The role of EGF repeats outside LBD and *O*-glycosylation on them for the activation of *Drosophila* Notch. EGF1-5 are dispensable for the activation of *Drosophila* Notch as well [67]. EGF24-26 are indispensable for the SER-induced activation [91]. The introduction of an *O*-fucose site on EGF14 (split) makes Notch more sensitive to DL [103]. Xylosyl-extension of *O*-glucose glycans on EGF16-20 are shown to inhibit the DL-induced activation while maintaining SER-induced activation [104,105,106]. Only the key sugar residues described in the main text are depicted. Sugar symbols are the same as in Figure 1C.

**Table 1 biomolecules-11-00309-t001:** A summary of the reported effects of *O*-glycosylation on Notch-ligand binding and Notch activation.

NOTCH1/DLL1	Ligand Binding	Activation
+Fringe	Increased	Increased
Thr466Val (*O*-Fucose site on EGF12)	Decreased	Decreased
Thr466Val + Ser435Ala (*O*-Glucose site on EGF11)	Nearly equal to Thr466Val	Further Decreased
**NOTCH1/JAG1**	**Ligand Binding**	**Activation**
+Fringe	Increased	Decreased

## Data Availability

No new data were created or analyzed in this study. Data sharing is not applicable to this article.

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
