# Peer review of "Current Views on the Roles of *O*-Glycosylation in Controlling Notch-Ligand Interactions"

_biomolecules, 2021, doi:10.3390/biom11020309_

Round 1

Reviewer 1 Report

This is a very detailed and well documented review on the roles of O-glycosilation in Notch-ligand interactions

Only limited concerns should be addressed as follows:

Both the "Abstract" and the "Perspective" sections outline the potential importance of  the Notch signaling in human development and diseases, as well as in the creation of new therapeutic strategies in various contexts, based on the modulation  of Notch signaling via Notch glycosylation. However, the remaining parts of the review do not develop these important aspects, especially the role attributed to the Notch signaling in human diseases that should be better outlined, at least in the Introduction

The text is on the whole well written and the language appears correct, however, an effort to synthesize the text would be appreciated. Some sections should be shortened to better focus on the contents. On the other hand  a paragraph of conclusions summarizing the most relevant  information would facilitate the reading also for not super-specialized  readers.

Author Response

Reviewer 1

This is a very detailed and well documented review on the roles of O-glycosilation in Notch-ligand interactions

Only limited concerns should be addressed as follows:

Both the "Abstract" and the "Perspective" sections outline the potential importance of the Notch

signaling in human development and diseases, as well as in the creation of new therapeutic strategies in various contexts, based on the modulation of Notch signaling via Notch glycosylation. However, the remaining parts of the review do not develop these important aspects, especially the role attributed to the Notch signaling in human diseases that should be better outlined, at least in the Introduction

We would like to thank the reviewer for the careful and positive evaluation of our manuscript. In response to the first point that the reviewer made, we added a concise summary of the role of Notch signaling in human diseases at the end of the first paragraph in the Introduction as follows.

In humans, dysregulation of Notch signaling leads to numerous diseases ranging from

developmental syndromes to adult-onset disorders. Mutations of Notch pathway components cause monogenic diseases such as Alagille syndrome and spondylocostal dysostosis.

Tumorigenesis is also related to dysregulation of Notch signaling in many cellular contexts, in which the Notch signaling pathway act oncogenic function such as in T-ALL as well as tumor suppressor function such as in the small cell lung cancer.

The text is on the whole well written and the language appears correct, however, an effort to synthesize the text would be appreciated. Some sections should be shortened to better focus on the contents. On the other hand a paragraph of conclusions summarizing the most relevant information would facilitate the reading also for not super-specialized readers.

We would like to thank the reviewer for the great suggestion. This is related to the answer to the second reviewer, but in the revised version, to better focus on the theme in this article, we deleted the original Section 3-6 and all the related text in other places. This is because the involvement of O-glycans in the interaction between receptors and ligands is the subject of this paper, but the involvement of O-glycans in the processes after ligand binding described in Section 3-6 is still vague and speculative, although it is an interesting possibility.

Here, we outline the current understanding of how Notch and its ligand interactions are

regulated by O-glycosylation. The function of the O-glycans is site-specific, in other words, which EGF repeats are attached to the glycans is also important for the function of the O-glycans. In particular, O-fucose glycans appear to function both directly and indirectly in the recognition of receptors by ligands. To clarify the molecular mechanism by which O-glycosylation of EGF repeats other than the ligand-binding region is involved in the interaction, it will be necessary to reveal the overall structure of the receptor including the O-glycans.

Reviewer 2 Report

The manuscript “Current views on the roles of O-glycosylation in controlling Notch-ligand interactions” by Saiki et al gives a thorough summary of O-glycosylation and its role in Notch signaling.  Overall, it is well written and reasonably easy to understand.  Although several reviews have been published on this and closely related aspects of Notch signaling, the authors present this review from the point of view of how glycosylation affects the ligand/receptor interaction which distinguishes it from other reviews on the subject. 

Major comment: section 3.5 could benefit from a summary figure.  I do not feel Figure 4, especially panel 4b, adds significantly to the manuscript.

Minor comments:

  1. Pg 2 line 68. The wording here suggests the dorsal part of the wing imaginal disc is the only expression domain.  While the authors are correct that FNG is restricted to the dorsal portion of the wing imaginal disc, it is not exclusive to the wing other discs for instance also show FNG expression in the dorsal parts.  The wording is not technically correct, even though the idea the authors are citing is correct.
  2. Pg 5. Lin 154. The section entitled “2-2. Advances in the understanding of the recognition and modification of Notch EGF repeats by 154 enzymes” nicely summarizes the modification of Notch EGF repeats by Glycosyltransferases, but doesn’t talk as much about the recognition.  Only the statement “The data provides detailed mechanisms for the recognition of properly folded EGF repeats using these enzymes” (line 171) but does not elaborate.  Quite a bit is known about how the GT-A and GT-B motifs recognize EGF repeats and the contacts made between them, as the authors suggest.  Given the title of the section, maybe a couple of sentences that elaborate the data in the literature could be helpful here.
  3. Pg 5. Line 177: the author states that “The retaining mechanism of glycosyltransferases is currently being debated.”  Yet it is not clear what aspects are being debated.
  4. Regarding figure 4 and accompanying text (e.g. line 314). The data supporting the existence of an additional step between ligand binding and cleavage is presented extremely tentatively, making it unclear if anyone has proposed an alternative or does the data clearly support an additional step but the mechanism is just unknown?  Some discussion of the strength of the conclusion and alternative hypotheses would be helpful here.    The way this is written confuses this issue (for example line 314 seems to make it clear an additional step exists).  Additional discussion on pg14 sounds more convincing but is then in contrast with the wording here that calls into question the existence of an extra step.  Taken together, the data seem to show something unknown exists, but it is not very clear.  Figure 4B does not really add much value.
  5. on Pg. 12 line 420: “FIX EGF1” needs to be defined when using an abbreviation.

Author Response

Reviewer 2

The manuscript “Current views on the roles of O-glycosylation in controlling Notch-ligand

interactions” by Saiki et al gives a thorough summary of O-glycosylation and its role in Notch

signaling. Overall, it is well written and reasonably easy to understand. Although several reviews have been published on this and closely related aspects of Notch signaling, the authors present this review from the point of view of how glycosylation affects the ligand/receptor interaction which distinguishes it from other reviews on the subject.

Major comment: section 3.5 could benefit from a summary figure. I do not feel Figure 4, especially panel 4b, adds significantly to the manuscript.

We would like to appreciate the constructive comments on our manuscript. We generated a new figure representing Section 3-5 “Structure and function of O-glycosylation outside of the LBD” as new Figure 4 in the revised manuscript. Moreover, we deleted the original Figure 4 since we agreed with the reviewer that it does not add any weight to the manuscript.

Minor comments:

  1. Pg 2 line 68. The wording here suggests the dorsal part of the wing imaginal disc is the only

expression domain. While the authors are correct that FNG is restricted to the dorsal portion

of the wing imaginal disc, it is not exclusive to the wing other discs for instance also show

FNG expression in the dorsal parts. The wording is not technically correct, even though the

idea the authors are citing is correct.

We agree with the reviewer that the wording is not correct. We deleted the word “exclusively” and revised the sentence as follows.

“FNG is expressed in the dorsal, but not ventral, part of wing imaginal discs and inhibits

the responsiveness of NOTCH to SERRATE (SER) expressed in dorsal cells,”

  1. Pg 5. Lin 154. The section entitled “2-2. Advances in the understanding of the recognition and modification of Notch EGF repeats by 154 enzymes” nicely summarizes the modification of Notch EGF repeats by Glycosyltransferases, but doesn’t talk as much about the recognition. Only the statement “The data provides detailed mechanisms for the recognition of properly folded EGF repeats using these enzymes” (line 171) but does not elaborate. Quite a bit is known about how the GT-A and GT-B motifs recognize EGF repeats and the contacts made between them, as the authors suggest. Given the title of the section, maybe a couple of sentences that elaborate the data in the literature could be helpful here.

We would like to thank the reviewer for pointing out that we did not describe the issue

adequately. We revised the sentence and added several sentences about acceptor recognition by glycosyltransferases as below.

“In general, for GT-B fold glycosyltransferases, the geometry of the cleft between donor

binding subsite and acceptor binding subsite vary greatly among enzymes to accommodate different acceptors such as glycan and protein domain. The structural data provide detailed mechanisms for the recognition of properly folded EGF repeats using these enzymes. The large cleft of POFUT1 and POGLUT1 show high complementarity with the folded EGF acceptor substrates. For example, POGLUT1 recognizes common 3D features of the properly folded EGF repeats that have the characteristic kinked loop of O-glucosylation consensus sequence as well as the conserved hydrophobic residue apart from the modification site in a primary sequence.”

  1. Pg 5. Line 177: the author states that “The retaining mechanism of glycosyltransferases is currently being debated.” Yet it is not clear what aspects are being debated.

We thank the reviewer for making this point. We added more information about the issue as below.

“The reaction mechanism of most inverting glycosyltransferases is generally considered as a single displacement SN2 mechanism; however, the retaining mechanism of glycosyltransferases is currently being debated. Two mechanisms have been proposed. The double displacement mechanism was first suggested, however, since lacking experimental evidence, in recent years some researchers proposed the SNi-like (i for internal return) frontface mechanism which retains configuration by letting the acceptor nucleophile attack the donor anomeric carbon from the same side as the leaving group.”

  1. Regarding figure 4 and accompanying text (e.g. line 314). The data supporting the existence

of an additional step between ligand binding and cleavage is presented extremely tentatively,

making it unclear if anyone has proposed an alternative or does the data clearly support an

additional step but the mechanism is just unknown? Some discussion of the strength of the

conclusion and alternative hypotheses would be helpful here. The way this is written

confuses this issue (for example line 314 seems to make it clear an additional step exists). Additional discussion on pg14 sounds more convincing but is then in contrast with the wording here that calls into question the existence of an extra step. Taken together, the data seem to show something unknown exists, but it is not very clear. Figure 4B does not really add much value.

As we mentioned to the first reviewer and at the beginning of our response to this reviewer, we deleted the original Figure 4. Accordingly, we deleted the original section 3-6 and all the related text in other places. This is because the involvement of O-glycans in the interaction between receptors and ligands is the subject of this paper, but the involvement of O-glycans in the processes after ligand binding described in the original section 3-6 is still vague and speculative, although it is an interesting possibility. The numbering of the following sections was revised accordingly.

Instead, we generated a new Table 1 to show the information described in the original Figure 4A.

  1. on Pg. 12 line 420: “FIX EGF1” needs to be defined when using an abbreviation.

It is spelled out as coagulation factor IX EGF1 in the revised manuscript.